# ADCs and TCE in SCLC Therapy: The Beginning of a New Era?

**DOI:** 10.3390/curroncol32050261

**Published:** 2025-04-30

**Authors:** Paola Muscolino, Fausto Omero, Desirèe Speranza, Carla Infurna, Silvana Parisi, Vincenzo Cianci, Massimiliano Berretta, Alessandro Russo, Mariacarmela Santarpia

**Affiliations:** 1School of Specialization in Medical Oncology, Department of Human Pathology “G. Barresi”, University of Messina, 98125 Messina, Italy; paola.muscolino@studenti.unime.it (P.M.); fausto.omero@studenti.unime.it (F.O.); carla.infurna@studenti.unime.it (C.I.); 2Department of Chemical, Biological, Pharmaceutical and Environmental Sciences, University of Messina, 98125 Messina, Italy; desiree.speranza@studenti.unime.it; 3Radiation Oncology Unit, Department of Biomedical, Dental and Morphological and Functional Imaging Sciences, University of Messina, 98125 Messina, Italy; silvana.parisi@unime.it; 4Department of Biomedical and Dental Sciences and Morphofunctional Imaging, University of Messina, 98125 Messina, Italy; 5Division of Medical Oncology, AOU “G. Martino” Hospital, University of Messina, 98124 Messina, Italy; massimiliano.berretta@unime.it; 6Department of Clinical and Experimental Medicine, University of Messina, 98122 Messina, Italy; 7Department of Medical Oncology, Humanitas Istituto Clinico Catanese, 95045 Misterbianco, Italy; alessandro.russo@humanitascatania.it; 8Department of Biomedical Sciences, Humanitas University, 20072 Pieve Emanuele, Italy; 9Department of Human Pathology “G. Barresi”, University of Messina, 98125 Messina, Italy

**Keywords:** SCLC, ICIs, TCE, ADCs, immunotherapy, ifinatamab deruxtecan, sacituzumab govitecan, tarlatamab

## Abstract

The therapeutic landscape for small cell lung cancer (SCLC) has remained stationary for decades, with chemotherapy representing the sole treatment strategy, with a modest survival benefit. The addition of immune checkpoint inhibitors (ICIs) to standard first-line chemotherapy for SCLC was a considerable milestone. However, despite high overall response rates, this strategy failed to deliver long-term benefits for most patients, who continue to face a poor prognosis. Over the last few years, a deeper knowledge of the molecular biology of SCLC and the impressive advancements in drug development, have led to the generation of novel classes of systemic therapies that promise to revolutionize the current therapeutic scenario. Among the various therapeutic approaches in development, T-cell Engagers (TCE) and antibody-drug conjugates (ADCs) stand out due to their unique structural characteristics and mechanisms of action. These therapies represent a paradigm shift from traditional monoclonal antibody (mAb) and chemotherapy regimens, allowing direct engagement of multiple targets associated with tumor progression. In this review, we provide an overview of current drug development in SCLC, specifically focusing on these new agents, summarizing available evidence, and tracking future directions.

## 1. Introduction

### 1.1. SCLC Epidemiology and Molecular Classification

Lung cancer is the second most common cancer in the US, following prostate cancer in men and breast cancer in women. It is classified into small-cell (15%) and non-small cell (85%) types, the latter is divided into three main subtypes, including adenocarcinoma, squamous cell carcinoma, and large cell carcinoma [1]. Small cell lung cancer (SCLC) is a high-grade malignant epithelial tumor that arises predominantly in the lung. Rarely, small cell carcinomas with similar histological features may arise from extrapulmonary sites (e.g., gastrointestinal or genitourinary tract) and are classified as extrapulmonary small cell carcinomas (EPSCC). Tobacco smoking is the primary risk factor for pulmonary SCLC. SCLC is classified according to its stage in limited stage (LS-SCLC), confined to one hemithorax without distant metastasis, and extensive stage (ES-SCLC) [2].

Its prognosis is poor with 5-year survival rates of about 7% due to advanced-stage diagnosis in most cases and the disease’s aggressiveness [3,4]. Metastases can develop throughout the body but primarily affect the lung, liver, bone, adrenal glands, and lymph nodes; at diagnosis, brain metastases can be already present in 10% of patients but during the course of the disease this percentage increases to more than 50% [5].

The most common gene mutations in SCLC include *TP53*, ‘the guardian of the genome’, and *RB1*, which appears to be mutated in 95% of cases [6].

Over the last years, different classifications have been proposed for SCLC based on its molecular characteristics, and, more recently, extensive transcriptional analyses led to the identification of distinct molecular subtypes.

In 2019, Rudin et al. proposed a new molecular classification of SCLC to predict patient prognosis. The analyzed genes included achaete-scute homologue 1 (*ASCL1*; also known as *ASH1*) and neurogenic differentiation factor 1 (*NeuroD1*) which are involved in the maturation of lung neuroendocrine cells. *INSM1* is a gene associated with high expression of both *ASCL1* and *NeuroD1*. Some SCLC subtypes rarely showed *ASCL1* or *NeuroD1* mutations but rather expressed others, including those derived from the coding of *YAP1* and *POU2F3*, which are related to the loss of neuroendocrine characteristics of the tumor. The authors classified SCLC as SCLC-A, SCLC-N, SCLC-Y, and SCLC-P, respectively, based on the expression of *ASCL1*, *NeuroD1*, *YAP1*, and *POU2F3* [7]. Among the therapeutic implications of this classification, for example, SCLC subtypes with high levels of the *YAP* protein expressed other genes involved in immune checkpoints and changes in the tumor microenvironment towards an ‘inflamed’ type that is associated with immunotherapy sensitivity. For this reason, a new molecular classification in SCLC-A, SCLC-N, SCLC-I (inflamed) non-NE, and NE, in relation to the neuroendocrine characteristics of the tumor, has recently been proposed [8] (Figure 1).

### 1.2. Standard of Care

Small-cell lung cancer treatment depends largely on the disease stage. Limited-stage SCLC can be treated with a multimodal approach including chemotherapy and radiation therapy. Surgical resection is rarely performed, as it is usually reserved for very small tumors (T1a-b) without evidence of mediastinal lymph node involvement, with most of the patients receiving concurrent chemo-radiotherapy. Extensive-stage SCLC is treated with systemic therapies, including chemotherapy and immunotherapy. Several chemotherapeutic agents and schedules have been studied, but platinum-based chemotherapy has been the mainstay of treatment for both LS- and ES-SCLC, with the preferred regimens represented by cisplatin/carboplatin and etoposide [9].

The association between cisplatin and etoposide for SCLC treatment was initially studied in 1985, showing a high percentage of objective responses (including 43% of complete responses), thereby becoming the standard of care regimen [10].

Despite initial sensitivity to chemotherapy, almost all cases progress due to the development of resistance, limiting the long-term clinical benefit [11]. This prompted the evaluation of novel treatment strategies to improve survival outcomes. Recently, the therapeutic armamentarium of both limited and extensive-stage SCLC has been improved with the introduction of Immune Checkpoint Inhibitors (ICIs), changing the standard of care after three decades of failures. Principal immunotargets are the receptor PD-1 (Programmed Cell Death Protein 1) with its ligand PD-L1 and CTLA-4 (Cytotoxic T Lymphocyte-Associated Antigen-4) with its ligand CD80. CTLA-4-CD80 binding results in T-cell inhibition. PD-1/PD-L1 pathway also negatively modulates the activity of T lymphocytes and of other cells involved in both innate and adaptive immunity. Hence, the use of anti-CTLA-4 and anti-PD-1/PD-L1 antibodies restores the anti-tumor activity of T cells [12]. The combination of ICIs and chemotherapy has demonstrated superior efficacy compared to chemotherapy alone.

In the phase 3 IMpower133 study, patients received atezolizumab plus carboplatin-based chemotherapy for four cycles (induction phase) followed by atezolizumab alone (maintenance phase) until unacceptable toxicity or progression of disease. Median overall survival (mOS) in the experimental arm was 12.3 months compared to 10.3 months in the standard arm (*p* = 0.007), while median progression-free survival (mPFS) was 5.2 months and 4.3 months (*p* = 0.02), respectively [13]. Interestingly, the 5-year OS data of IMpower133 and the IMbrella A extension study revealed that a small, but clinically significant proportion of patients with ES-SCLC can derive long-term benefits from PD-L1 blockage, with 12% of patients still alive at 5 years [14]. Similarly, the phase 3 CASPIAN trial studied the combination of another PD-L1 inhibitor, durvalumab, in combination with standard platinum-based chemotherapy (cisplatin or carboplatin plus etoposide). Subjects were randomized to receive chemo-immunotherapy for up to six cycles, followed by maintenance durvalumab or chemotherapy alone or chemoimmunotherapy with durvalumab and tremelimumab, an anti-CTLA-4 antibody. Immunotherapy addition resulted in OS prolongation (13.0 months vs. 10.3 months in the platinum-etoposide group, with 34% vs. 25% of patients alive at 18 months). Tremelimumab plus durvalumab arm did not improve the mOS compared to chemotherapy alone. Thirty-six-month OS rate was 17.6% versus 5.8% respectively in the experimental and control arms [15,16].

Based on these results, FDA and EMA approved chemo-immunotherapy combinations with atezolizumab or durvalumab as first-line treatment for ES-SCLC. Despite the small but clinically significant improvement in OS, the use of chemo-immunotherapy in ES-SCLC is burdened by the absence of predictive biomarkers that can identify patients that will derive benefit from this strategy [17]. Currently, no specific biomarker is approved for chemo-immunotherapy in this disease and all the biomarkers tested to date have failed to predict any significant benefit from ICIs. For example, SCLC is characterized by a high tumor mutational burden (TMB) that in other tumors is associated with an increased release of tumor antigen resulting in a better response to immunotherapy; however, TMB response predictivity in both tissue and plasma (bTMB) in SCLC appears reduced [13]. Similarly, PD-L1 expression was not predictive of therapeutic efficacy [18].

In case of SCLC relapse, patients may receive a second-line treatment, depending on the time to relapse. Early SCLC relapse, within six months, might benefit from topotecan, lurbinectedin, or other chemotherapy regimens. Topotecan has been compared to a CAV chemotherapy schedule (cyclophosphamide, doxorubicin, and vincristine) in SCLC with early relapse. Median times to progression and median survival were similar in both arms but topotecan monotherapy reduced chemotherapy-related symptoms and the rate of adverse events [19]. At the same time, lurbinectedin has been evaluated as a second line, in a phase 2 study, in pre-treated patients with platinum-based chemotherapy, showing a promising activity in terms of overall response and tolerability [20]. Several chemotherapeutic agents are recommended in second-line SCLC therapy, such as paclitaxel, temozolomide, docetaxel, irinotecan, vinorelbine, and gemcitabine. In addition, patients with late relapse (>6 months) may benefit from platinum/etoposide rechallenge [9]. Patients with metastatic SCLC may also receive thoracic and/or encephalic radiotherapy (RT). Thoracic RT is an option in case of complete response of distant metastasis and residual intrathoracic disease. There were no significant differences in terms of toxicity and tolerability in radio-treated patients. Thoracic RT, in ES-SCLC, chemo-treated, improved survival rates compared with chemotherapy alone in the pre-immunotherapy era [21,22] but seems to improve OS even after chemo-immunotherapy and ongoing studies will further clarify its role [23].

SCLC tends to metastasize to the central nervous system (CNS). For this reason, in patients with good performance status, encephalic RT is recommended and may reduce CNS metastases growth. On the other hand, prophylactic cranial irradiation (PCI) can lead to the development of neurocognitive decline, especially in elderly subjects [24,25].

Given the limited success of traditional chemotherapy and immunotherapy, innovative therapeutic modalities such as T-cell Engagers (TCEs) and antibody-drug conjugates (ADCs) have gained increasing attention.

## 2. T-Cell Engagers and Antibody-Drug Conjugates: Structure and Function

### 2.1. TCE

T-cell Engagers are a class of engineered molecules designed to target two different molecules simultaneously. This dual-targeting approach creates a direct link between T cells and tumor cells, thereby facilitating an immune response against cancer [26]. A typical TCE consists of two key structural components: single-chain variable fragments (scFvs) and flexible linker regions that connect them [27]. The design of the TCE allows it to effectively recruit T cells to the tumor site, which is particularly beneficial in cancers such as SCLC, where immune evasion mechanisms often hinder the body’s natural immune response to tumors. The scFvs, derived from the variable regions of antibodies, exhibit distinct binding specificities to various target antigens. One scFv is designed to bind to a tumor-associated antigen, while the other is engineered to attach to CD3, a component of the T-cell receptor complex (Figure 2). By linking T cells and tumor cells, this dual-targeting approach enhances immune-mediated tumor killing [28]. The scFvs are linked by a short peptide, which is essential to maintain the flexibility required for effective binding. The function of this linker is to ensure that both scFvs can independently connect to their respective targets, forming a functional immunological synapse between T cells and tumor cells. The length and composition of the linker are critical in determining the flexibility and activity of the TCE because they must allow both scFvs to efficiently reach their targets while maintaining stability [29]. Some TCEs are engineered to include an Fc region, derived from the constant region of antibodies, to improve pharmacokinetics, stability, and potential immune interactions. While this Fc region can enhance antibody-dependent cellular cytotoxicity (ADCC) and immune activation, it may also contribute to side effects, such as cytokine release syndrome (CRS) in certain patients [29].

TCEs act by first activating T cells through CD3 targeting. This interaction triggers the secretion of cytokines such as interferon-gamma and interleukin-2, which amplify the immune response. T-cells then release cytotoxic molecules, such as perforin and granzymes, which induce apoptosis in tumor cells. The small size of scFvs allows for better tissue penetration and efficient biodistribution, which is particularly crucial for delivering therapeutic agents to deep tumors, such as those found in SCLC [29].

### 2.2. ADCs

Antibody-drug conjugates represent a sophisticated class of targeted cancer therapies that combine the specificity of monoclonal antibodies (mAbs) with the potent cytotoxic effects of chemotherapeutic agents [30]. ADCs consist of three primary structural components: the mAb, the linker, and the cytotoxic payload. The mAb acts as a highly specific targeting agent, binding to a specific antigen often overexpressed on the surface of cancer cells. This antigen, known as tumor-associated antigen (TAA), can be surface-expressed or internalized by cancer cells. The antibody structure includes two variable regions (Fab regions) responsible for antigen binding and two constant regions (Fc regions) that facilitate immune activation. The antibody is often modified to optimize pharmacokinetics, stability, and internalization into the target cells to ensure effective delivery [31].

The cytotoxic drug or payload is a potent chemotherapeutic or other cytotoxic agent—such as a maytansinoid, auristatin, calicheamicin, or DNA damaging agents—that kills cancer cells once inside. These drugs are too toxic to be administered systemically in their free form, so they are delivered via ADCs to minimize off-target toxicity. The linker, which binds the mAb to the cytotoxic payload, plays a critical role in ensuring that the drug remains stable in circulation and prevents premature release. Once inside the target cancer cell, the linker is designed to be labile, releasing the drug at the correct site. Linkers can be cleavable—by enzymes such as cathepsins or by reductions in the intracellular environment—or non-cleavable, where the drug is released only upon antibody degradation. The ligand, as the central component of ADCs, serves as the bridge that connects the antibody to the cytotoxic drug (Figure 3)

Selecting a suitable ligand is essential because it must remain stable during circulation to ensure the drug is not prematurely released before reaching the tumor site [32]. This stability minimizes off-target effects, which could otherwise cause adverse side effects. In addition, the ligand must be designed to cleave within the tumor microenvironment or after internalization to facilitate the effective release of the cytotoxic drug. Cytotoxic agents vary widely between ADCs and can be categorized as microtubule inhibitors, DNA-damaging agents, and other novel compounds. These agents are all selected for their ability to induce apoptosis in targeted cancer cells [33]. For example, auristatins and maytansinoids disrupt microtubule dynamics, while agents such as calicheamicin induce double DNA strand breaks, ultimately leading to cell death. ADCs exert their therapeutic effects through a systematic process that begins with the specific binding of the ADC to tumor-associated antigens on the surface of cancer cells. The high affinity of the mAb ensures that it recognizes and binds to specific markers overexpressed on cancer cells, such as CD56, CD133, or NSE in SCLC [34].

Once bound, the ADC is internalized via receptor-mediated endocytosis, where the cancer cell membrane engulfs the ADC and delivers it to the cytoplasmic compartment. Once inside the cell, the ADC dissociates in the acidic environment of endosomes or lysosomes, releasing the cytotoxic payload, which disrupts key cellular processes and induces cell cycle arrest, leading to apoptosis [35]. The specificity of the antibody ensures that the cytotoxic effects are primarily confined to cancer cells, minimizing damage to normal tissue—a significant advantage over conventional chemotherapy, which lacks this level of targeting.

### 2.3. ADCs and TCEs in SCLC

#### 2.3.1. Delta-Like Ligand 3 (DLL3)

Delta-like ligand 3, normally localized intracellularly, inhibits Notch signaling but is aberrantly expressed on the surface of SCLC cells. Across the different molecular subtypes, the expression of DLL3 is variable [36]. Tarlatamab is a TCE that directs the patient’s T cells to cancer cells expressing DLL3, independent of major histocompatibility complex (MHC) class I. The Tarlatamab mechanism of action is based on its ability to simultaneously bind to T cells via CD3 and to tumor cells via DLL3. CD3 is a component of the T cell receptor (TCR) complex, which plays a key role in T cell activation and immune response. DLL3, a membrane-bound protein of the Notch ligand family, is aberrantly expressed in several cancers, including SCLC. The binding of tarlatamab to CD3 leads to cross-linking of T cells and tumor cells, bringing cytotoxic T cells into close proximity with tumor cells. This proximity facilitates the immune-mediated destruction of tumor cells. At the same time, tarlatamab binds to DLL3 on the surface of tumor cells, directing the immune response specifically to cancer cells that express this protein [37]. Engaging T cells by CD3 activates the TCR complex, which triggers a cascade of intracellular signaling events. These events lead to T-cell proliferation and the release of cytokines. Activated T cells then gain the ability to directly target and destroy tumor cells. They release cytotoxic molecules such as perforin and granzymes, that work together to induce apoptosis in tumor cells. Perforin creates pores in the target cell membrane, allowing granzymes to enter the tumor cells and initiate a series of proteolytic events that lead to cell death [38].

##### DeLLphi-300

Tarlatamab was first evaluated in a phase I trial (DeLLphi-300) assessing the safety, activity, and pharmacokinetics of this agent in patients with relapsed/refractory SCLC. The study enrolled 107 patients heavily pretreated (median prior anticancer therapies 2, range 1–6, and ~50% had previous PD-1/PD-L1 exposure) who had received tarlatamab in dose exploration (0.003 to 100 mg; *n* = 73) and expansion (100 mg; *n* = 34) cohorts. Maximum tolerated dose (MTD) was not reached, with 90.7% of patients experiencing any adverse events (AEs) (30.8% grade ≥ 3, and 1% G5 pneumonitis).

AEs of special interest were (CRS), occurring in 56 patients (52%) including grade 3 in one patient (1%). Preliminary activity was promising overall response rate (ORR 23.4%), and duration of response (DoR 12.3 months) [39]. An exploratory analysis of patients with brain metastases suggested encouraging intracranial activity of tarlatamab with CNS tumor shrinkage of ≥30% was observed in 62.5% of patients who had a baseline CNS lesion of ≥10 mm (*n* = 16) [40]. The impact of tarlatamab activity on brain metastases is not fully understandable, as all the patients had received brain radiotherapy before study entry.

##### Dellphi-301

The phase 2 trial, DeLLphi-301, further explored the antitumor activity and safety of tarlatamab at two different doses (intravenously every 2 weeks at a dose of 10 mg or 100 mg) in pretreated SCLC patients with prior two or more lines of therapy [41].

An objective response occurred in 40% (97.5% CI, 29 to 52) of the patients in the 10-mg group and in 32% (97.5% CI, 21 to 44) of those in the 100-mg group. The median PFS was 4.9 months (95% CI, 2.9 to 6.7) in the 10-mg group and 3.9 months (95% CI, 2.6 to 4.4) in the 100-mg group. Indeed, tarlatamab 10 mg was associated with activity in both patients with stable and treated brain metastases (*n* = 23) and without brain metastases (*n* = 77): ORR 52% vs. 38%, median PFS 6.7 (CI 95%, 3-NE) months vs. 4.0 months (CI 95%, 3–6), and median OS 14.3 months (CI 95%, 14-NE) vs. not estimable (9-NE), respectively. Interestingly, in patients with brain metastases ≥10 mm (3 patients in the 10 mg group and 14 patients in the 100 mg group), a CNS shrinkage ≥30% was observed in 59% of the cases with an intracranial control of 94% and a median duration of intracranial disease control not estimable (range 2.6–13.9+ months) [42], albeit this analysis is limited by small sample size. The most common AEs were CRS (in 51% of the patients in the 10-mg group and in 61% of those in the 100-mg group) and immune effector cell–associated neurotoxicity syndrome (ICANS) that included confusion, impaired attention, tremor, and motor findings. ICANS and associated neurologic events occurred more frequently in the higher dose group (8% in the 10-mg group and 28% in the 100-mg group), including grade ≥3 events (0% vs. 5%, respectively). Most of these events were observed during the first cycle, with a median time to onset of 5 days. OS data was still immature, at the last follow–up (~10 months), with 57% of patients in the tarlatamab 10 mg group and 51% of patients in the tarlatamab 100 mg group still alive [41].

Based on DeLLphi-301 trial data, on 16 May 2024, the FDA granted accelerated approval to tarlatamab for ES-SCLC after platinum-based chemotherapy.

##### Other Trials

Recently, an interim analysis of the DeLLphi-304 trial showed a statistical improvement in OS compared to standard chemotherapy (topotecan, lurbinectedin, or amrubicin) in patients progressing after platinum-based chemotherapy [43].

Multiple ongoing studies are evaluating tarlatamab in different settings, including combination with 1st line chemo-immunotherapy (phase 1b, DeLLphi-303), maintenance after chemo-immunotherapy (phase 3, DeLLphi-305), and subcutaneous formulation (phase 1b, DeLLphi-308) in ES-SCLC and after chemoradiotherapy (phase 3, DeLLphi-306) in LS-SCLC (Table 1).

Recent studies have also confirmed the benefit of using other antibodies directed against DLL3 in SCLC. Among these new molecules, BI 764532 (Obrixtamig) must be mentioned which showed increased ORR and estimated PFS across all dose levels. The most common AEs were CRS, dysgeusia, pyrexia, and asthenia [44].

Moreover, MK-6070 showed promising efficacy in SCLC, including in patients with brain metastases [45].

**Table 1 curroncol-32-00261-t001:** Trials with tarlatamab in SCLC.

Trial Name	Phase Trial	Setting	Drug	n	Primary Endpoint
DeLLphi-300	Phase I	Second line—ES	Tb	269	Safety [39]
DeLLphi-301	Phase II	Third or next line—ES	Tb	222	Safety, ORR [42]
DeLLphi-302	Phase Ib	Second line	Tb + IO	23	Safety [46]
DeLLphi-303	Phase Ib	First-line- ES	Tb + ChT + IO or Tb + IO	269	Safety [47]
DeLLphi-304	Phase III	Second line—ES	Tb	509	OS [43]
DeLLphi-305	Phase III	Maintenance—ES	Tb + IO		OS [48]
DeLLphi-306	Phase III	Maintenance—LS	Tb	400	PFS [49]
DeLLphi-307	Phase IIa	Third or next line—ES (Asiatic)	Tb	32	ORR [50]
DeLLphi-308	Phase Ib	ES	SC Tb	100	Safety [51]
DeLLphi-309	Phase II	Second line	Tb	240	ORR [52]

Abbreviations: extensive stage (ES), limited stage (LS), tarlatamab (Tb), chemotherapy (ChT), immunotherapy (IO), subcutaneous (SC), overall survival (OS), overall response rate (ORR), progression-free survival (PFS).

#### 2.3.2. Trophoblast Surface Antigen 2 (TROP-2)

In addition to DLL3-targeted therapies, TROP-2-targeted ADCs have also demonstrated promising efficacy in SCLC. TROP-2 is a transmembrane glycoprotein and a calcium signaling transducer expressed in many epithelial tumors, including SCLC with a high expression observed in 10% of patients [8].

##### TROPiCS-03

Sacituzumab govitecan (SG) is a first-in-class ADC directed at TROP-2, composed of a humanized mAb binding to SN-38, the active metabolite of the topoisomerase I inhibitor irinotecan. SG is approved worldwide for patients with pre-treated, triple-negative, or hormone receptor-positive metastatic breast cancer with human epidermal growth factor receptor 2 negative. The open-label, multicohort, phase 2 TROPiCS-03 study (NCT03964727) evaluated SG in patients with metastatic or locally advanced solid tumors, including ES-SCLC, showing promising activity. SG was administered via intravenous infusion at 10 mg/kg on days 1 and 8 of each 21-day cycle (until PD or unacceptable toxicity) in 43 pretreated ES-SCLC. ORR was 41.9% (95% CI, 27.0–57.9); the DOR rate at 6 months was 48.2% (95% CI, 23.9–68.9). Median PFS was 4.40 months (95% CI, 3.81–6.11) and median OS was 13.60 months (95% CI, 6.57–14.78). SG demonstrated antitumor activity in patients with both platinum-resistant (ORR, 35.0%; 95% CI, 15.4–59.2) and platinum-sensitive (ORR, 47.8%; 95% CI, 26.8–69.4) disease. Regarding safety, all patients presented with ≥1 treatment-emergent adverse events (TEAEs) of any grade, most commonly diarrhea (76.7%), fatigue (60.5%) and neutropenia (55.8%). The most common grade ≥3 TEAEs were neutropenia (44%) and diarrhea (9%) [53].

##### Other Trials

Another TROP-2 inhibitor is SHR-A1921, which is composed of a humanized anti-TROP-2 IgG1 mAb attached to DNA topoisomerase I inhibitor (payload, DXh) via a tetrapeptide-based cleavable linker (GGFG). A phase 1 study (NCT05154604) evaluated the efficacy and safety of SHR-A1921 in patients with pretreated ES-SCLC (*n* = 17). The dose is 3.0 mg/kg i.v. Every three weeks was selected. An encouraging activity was observed in this heavily pretreated population: ORR 33.3% (95% CI 15.2–58.3), median DoR 4.4 months (95% CI 2.3-NR), and median PFS 3.8 months (95% CI 1.4-NR). The most common TRAEs of any grade that occurred were stomatitis, vomiting, nausea, anemia, and decreased appetite [54].

#### 2.3.3. B7 Homolog 3 Protein (B7-H3)

B7 homolog 3 protein (also known as CD276) is a member of the B7 family overexpressed in tumor tissues and is a costimulatory/coinhibitory immunoregulatory protein that performs a dual role in the immune system during T-cell activation and is an emerging target for antibody-based immunotherapy [55]. In SCLC, B7H3 expression is high and remarkably consistent across the 4 distinct molecular subtypes and is not correlated with PD-L1 expression or T cells, but it has been reported a strong correlation with some immune checkpoint genes, including *HAVCR2/TIM3*, *CD86*, and *PDCD1LG2/PD-L2*, as well as M2 macrophages [36].

Ifinatamab deruxtecan (I-DXd) is a B7-H3–directed ADC composed by a humanized anti–B7-H3 IgG1 mAb, a tetrapeptide-based cleavable linker and a topoisomerase I inhibitor payload (an exatecan derivative, DXd). The mechanism of action of ifinatamab in SCLC begins with its highly targeted binding to B7-H3. Following binding, ifinatamab is internalized into the cancer cell by receptor-mediated endocytosis, where it enters endosomes or lysosomes. In these acidic compartments, the linker connecting the mAb to the cytotoxic payload is cleaved, releasing the potent cytotoxic drug inside the cancer cell. The released drug then determines the inhibition of topoisomerase I, an enzyme essential for inducing relaxation of the DNA double helix and its transcription. One of the key advantages of ifinatamab is its ability to bypass common resistance mechanisms seen with conventional chemotherapy. SCLC cells often develop resistance to traditional chemotherapeutic agents through mechanisms such as drug efflux pumps or mutations in drug targets, but ifinatamab’s targeted delivery allows it to bypass these resistance pathways, increasing its efficacy in resistant SCLC tumors. In addition, the Fc region of ifinatamab can engage immune cells such as natural killer cells and macrophages through ADCC or complement-dependent cytotoxicity, further enhancing its anti-tumor activity [56].

##### IDeate-Lung01, IDeate-Lung02 and IDeate-Lung03

The phase 2 study IDeate-Lung01 evaluated the activity and safety of I-DXd in pretreated ES-SCLC, which were stratified in two different groups (8 mg/kg and 12 mg/kg every 3 weeks), based on chemotherapy-free interval < or ≥ 90 days, the third or fourth line of therapy and prior or not anti–PD-(L)1 exposure. I-DXd showed higher ORRs in patients treated with 12 mg/kg (*n* = 42) vs. 8 mg/kg (*n* = 46): ORR 54.8% (95% CI, 38.7–70.2) and 26.1% (95% CI, 14.3–41.1), respectively. Median PFS and median OS were 4.2 months (2.8–5.6) and 9.4 months (7.8–15.9) for the 8 mg/kg arm and 5.5 months (4.2–6.7) and 11.8 months (8.9–15.3) for the 12 mg/kg arm, respectively.

The most common TEAEs were nausea (50% and 28%), decreased appetite (43% and 17%), and anemia (36% and 13%) for 12 mg/kg and 8 mg/kg doses, respectively. Treatment-related interstitial lung disease occurred in 4 and 5 patients treated with 8 and 12 mg/kg, respectively [57].

I-DXd is under evaluation in a phase 1/2 study (NCT04471727) in combination with the DLL-3 inhibitor MK-6070 (HPN328), in a phase 3 trial (IDeate-Lung02) at the dose of 12 mg/kg versus Treatment of Physician’s Choice in relapsed SCLC, and in a phase 1/2 study (IDeate-Lung03) in combination with 1st line chemo-immunotherapy in ES-SCLC.

##### Other Trials

Another B7-H3-targeted ADC is HS-20093 (GSK5764227) composed of a fully humanized anti-B7-H3 mAb covalently linked to topoisomerase I inhibitor payload (an exatecan derivative) via a cleavable maleimide tetrapeptide linker. The phase 1 study ARTEMIS-001 evaluated HS-20093 in the ES-SCLC cohort with dose-escalation and dose-expansion stages. Promising activity was observed, reporting an ORR of 61.3% (CI 95%, 42.2–78.2) in 8.0 mg/kg cohort (*n* = 31) and 50.0% (CI 95%, 28.2–71.8) in 10.0 mg/kg cohort (*n* = 22). Median PFS and median OS were 5.9 months (CI 95%, 4.4–8.5) and 9.8 months (95% CI: 8.5, NA) in 8.0 mg/kg cohort and 7.3 (CI 95%, 3.4–11.0) and not reached in 10.0 mg/kg cohort. A higher ORR was reported in patients with previous exposure to platinum plus immunotherapy therapy and no history of topoisomerase I inhibitor at either dose level. Clinical toxicities of HS-20093 mainly included hematologic and gastrointestinal events. Grade ≥3 TRAEs (with incidence ≥10%) were neutropenia (39.3%), thrombocytopenia (17.9%), and anemia (16.1%) [58]. Ongoing studies with HS-20093 include the phase 3 trial ARTEMIS-008 in relapsed SCLC compared with topotecan and the phase 3 ARTEMIS-009 as consolidation after chemo-radiotherapy in LS-SCLC.

#### 2.3.4. Seizure-Related Homolog Protein 6 (SEZ 6)

##### Seizure-Related Homolog Protein 6 Is a New Target Expressed in SCLC

ABBV-011 is a novel ADC targeting SEZ6. It includes the SC17 anti-SEZ6 mAb, conjugated with DNA-damaging payload N-acetyl-γcalicheamicin, via a new non-cleavable linker with a drug-to-antibody ratio of 2. ABBV-011 is aimed to release the calicheamicin payload intracellularly to induce cell death. The open-label phase I study of ABBV-011 was administered as monotherapy or in combination with budigalimab, a PD-1 inhibitor, in patients with relapsed/refractory SCLC. The first part of the study evaluated ABBV-011 as monotherapy and consisted of a dose-escalation phase followed by a dose-expansion phase. The second part of the study assessed the combination of ABBV-011 and budigalimab in dose-escalation and dose-expansion phases. Eligible patients included adults (≥18 years of age) with histologically or cytologically confirmed relapsed/refractory SCLC who received ≤3 Lines of prior therapy. ABBV-011 was administered intravenously once every 3 weeks during dose escalation (0.3–2 mg/kg) and expansion. In the 1-mg/kg dose-expansion cohort (n: 40), ORR was 25%; the median response duration was 4.2 months (95% CI, 2.6–6.7); and mPFS was 3.5 months (95% CI, 1.5–4.2) (Table 2).

The most common treatment-emergent adverse events were fatigue (50%), nausea (42%), and thrombocytopenia (41%). The most common hepatic treatment-emergent adverse events were increased aspartate aminotransferase (22%), increased γ-glutamyltransferase (21%), and hyperbilirubinemia (17%) [59]. SEZ6 is a promising new target for SCLC and will be the subject of future studies.

## 3. TCEs vs. ADCs: Therapeutic Scenarios and Challenges

Several studies have shown that new weapons against SCLC may be available in the near future. The mechanisms of action of these new drugs, as explained, are very different: on one hand, they can activate the immune system against tumor cells, and alternatively, they enable direct tumor delivery of potent chemotherapeutic agents, reducing toxicity and improving therapeutic responses.

In DeLLphi-301, tarlatamab showed excellent results in terms of duration of response to treatment but 97.7% of subjects experienced adverse events. Although the drug’s mechanism is immunostimulatory, the toxicities observed in the study differ from those typical of anti-PD-1/PD-L1 therapy. The most notable finding was that 51% of patients receiving 10 mg of tarlatamab developed CRS within the first hours after infusion, most events during the first cycle. Although this toxicity can be challenging to manage, no fatal events or treatment discontinuations due to this AE were reported [42].

The toxicity related to SG is already well known from its widespread use in breast cancer. The most common AEs were diarrhea, fatigue, and neutropenia. Although these events might appear more manageable, 7% of patients experienced fatal events, 37% required dose reduction, and 69% discontinued treatment [53].

The AEs related to the ADCs discussed in this article are primarily gastrointestinal and hematologic. However, there have also been rare cases of interstitial pneumonitis, already known to occur with the use of deruxtecan in breast cancer [56].

It will also be interesting to observe, over the long term, the use of anti-SEZ6 in combination with innovative chemotherapeutic agents such as N-acetyl-γcalicheamicin, and to evaluate their long-term toxicity [59].

The currently available results regarding the use of these new molecules in SCLC derive from trials conducted in the extensive stage setting and in lines of therapy beyond the first. A significant step forward in the development of innovative treatments will be the evaluation of ongoing trials, such as the combination of tarlatamab with first-line chemoimmunotherapy or its use in the maintenance phase. It will also be necessary to define the optimal therapeutic sequence and the potential use of these agents in limited-stage disease [43,44,45,46,47,48,49,50,51,52].

More scientific research on the molecular subgroups of SCLC may help to understand which patients can benefit most from treatment with TCEs and ADCs. For instance, it may be interesting to assess whether the SCLC-I subtype—associated with high YAP levels and the expression of genes involved in immune response modulation—responds better to TCEs than to ADCs. However, the currently available data remains immature.

## 4. Conclusions

After three decades of failures, PD-L1 inhibitors have changed the standard of care for the first-line systemic treatment of ES-SCLC and, more recently, of LS-SCLC, proving a small, but clinically relevant advantage in overall survival. However, no predictive biomarkers have been approved for these agents. The improved knowledge of the molecular pathology of SCLC has led to the identification of distinct molecular subtypes that may be associated with different sensitivity to current therapies and likely provide potential novel therapeutic opportunities. Additionally, novel classes of systemic therapies are rapidly emerging and promise to revolutionize further the therapeutic landscape of SCLC through their innovative mechanisms of action. Several ADCs targeting different highly expressed targets in SCLC and the new generation of immunotherapeutic agents, TCEs, are the most promising drugs, demonstrating encouraging durable anti-cancer activity and manageable safety. Future perspectives include the identification of the most effective therapeutic sequences, especially for immunotherapy, and the integration of the different therapeutic strategies with radiotherapy. Furthermore, other important factors include the development of tailored therapeutic approaches, the identification of reliable predictive biomarkers, and the potential use in early-stage disease.

## Figures and Tables

**Figure 1 curroncol-32-00261-f001:**
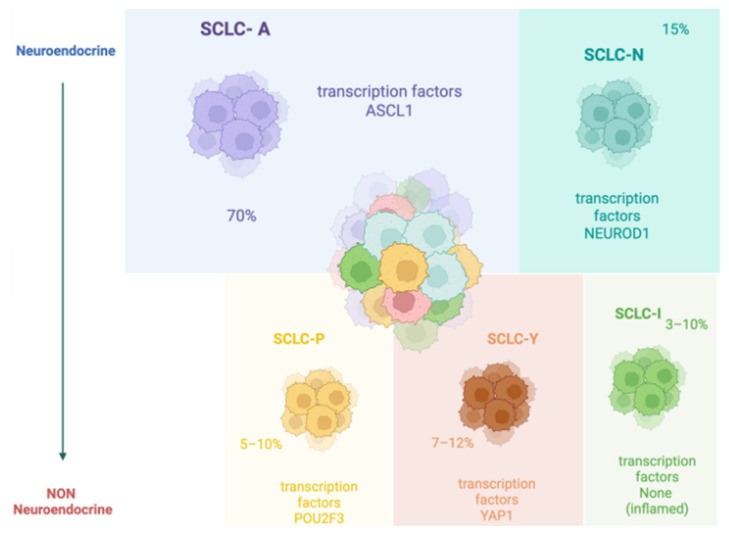
Molecular subtypes of Small cell lung cancer (SCLC) and major transcription factor with relative proportion to overall SCLC cases. (Credit: Created with BioRender.com).

**Figure 2 curroncol-32-00261-f002:**
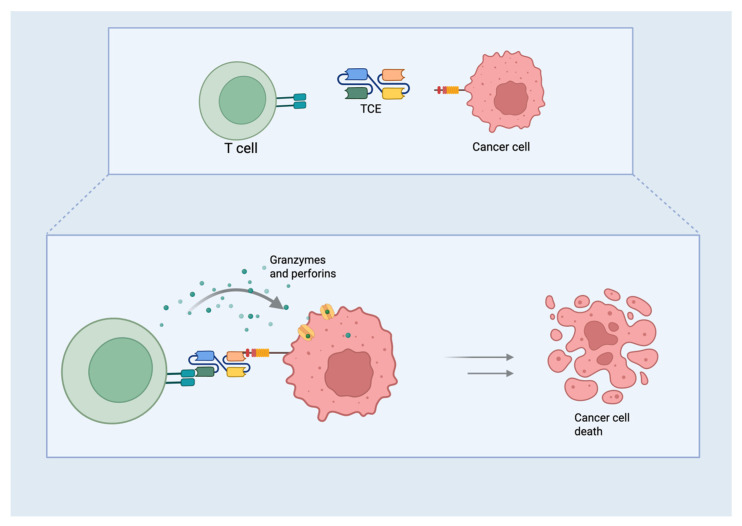
Structure and mechanism of action of TCEs. T-cell Engagers (TCE) antibody consists of several structural components: two single-chain variable fragments (scFvs) linked by a flexible linker region. The scFvs bind simultaneously to CD3 on T cells and to tumor-associated antigen overexpressed on the surface of SCLC cells. CD3 bond brings the T cells into proximity to the tumor cells, triggering T cell activation and cytotoxic activity. This leads ultimately to the tumor cells’ death via apoptosis (Credit: Created with BioRender.com).

**Figure 3 curroncol-32-00261-f003:**
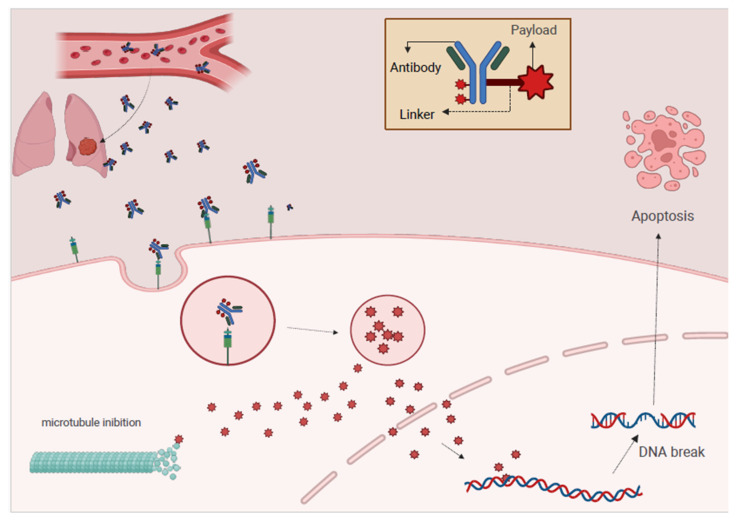
Structure and mechanism of action of ADC. Antibody-drug conjugates (ADCs) typically consist of: (i) a monoclonal antibody designed to specifically bind to a particular antigen overexpressed on the surface of tumor cells; (ii) a linker that covalently attaches the antibody to the cytotoxic drug; and (iii) a payload, which is a highly potent chemotherapeutic agent or toxin responsible for killing tumor cells. Once internalized, the linker is cleaved, and the payload is released. This causes DNA damage and microtubule disruption, ultimately leading to cell death via apoptosis (Credit: Created with BioRender.com).

**Table 2 curroncol-32-00261-t002:** Major ADCs in clinical development in pretreated SCLC.

Drug	Target	Trial Name	Setting	*n*	ORR (%)	mPFS (mos)	mOS (mos)	AEs G3-G5	Ref.
I-DXd	B7-H3	Phase 2, Ideate-Lung01	ES-SCLC	88	−12 mg/kg, 54.8% (95% CI, 38.7–70.2) −8 mg/kg, 26.1% (95% CI, 14.3–41.1)	−8 mg/kg, 4.2 mos (2.8–5.6) −12 mg/kg, 5.5 m (4.2–6.7)	−8 mg/kg, 9.4 mos (7.8–15.9) −12 mg/kg, 11.8 mos (8.9–15.3)	−43.5% (8 mg/kg) −50% (12 mg/kg)	[57]
HS-20093	B7-H3	Phase 1, ARTEMIS-001	ES-SCLC	56	−8.0 mg/kg, 61.3% (CI 95%, 42.2–78.2) −10.0 mg/kg 50.0% (CI 95%, 28.2–71.8)	−8 mg/kg, 5.9 mos −10.0 mg/kg, 7.3	−8 mg/kg, 9.8 mos −10.0 mg/kg, NR	N/A	[58]
SACITUZUMAB-GOVITECAN	TROP 2	TROPICS-03	ES-SCLC	43	41.9% (95% CI, 27.0–57.9)	4.40 mos (95% CI, 3.81–6.11)	12.2 mos	51.2%	[53]
SHR-A1921	TROP-2	Wang J et al., 2024	ES-SCLC	17	33.3% (95% CI 15.2-58.3)	3.8 mos (95% CI 1.4-NR)		35.3%	[54]
ABBV-011	SEZ6	(NCT03639194)	ES-SCLC	dose escalation, *n* 36; dose expansion, *n* 60	1-mg/kg dose-expansion cohort (*n* 40) 25%	3.5 mos (95% CI, 1.5–4.2)	N/A	34%	[59]

Abbreviations: progression-free survival (mPFS), overall response rate (ORR), adverse events (AEs), not available (N/A).

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
