# Peer review of "ADCs and TCE in SCLC Therapy: The Beginning of a New Era?"

_curroncol, 2025, doi:10.3390/curroncol32050261_

Round 1
Reviewer 1 Report
Comments and Suggestions for Authors
This review article is focused in new therapeutic approaches for small cell lung cancer (SCLC). The authors first describe the present therapeutic options and the need for improved treatments given the modest survival benefit of those that are available at the present time. The review is then centred in two new therapeutic approaches, biespecific T-cell engagers (BiTEs) and antibody-drug conjugates (ADCs). The mechanism of action of these new compounds is explained in detail. Later on, the authors described the results obtained for SCLC patients in some clinical trials, one of them for a BiTEs molecule and three for ADC combinations.
The review article is comprehensive and very well presented. It would be of interest for researchers in the field of SCLC and also in that of new cancer therapies involving activation of the immune system.
There are a few points where the review article could be improved, as follows:
- Using the same terminology along the manuscript could help to follow the article more easily. For example, biespecific T- cell engagers are named BiTEs in the title and Abstract but BiMAbs in the title of section 3 (line 247). In addition, the group of SCLCs with distant metastasis is named as “extensive disease (ED-SCLC) in the introduction (line 52) and as ES-SCLC at other places along the manuscript, like lines 86 and 427.
- The sentence in line 356 should be revised since it states that the ADC generated against B7-H3 binds to DLL3.
- The general sentence in lines 360-361: “The released drug then disrupts critical cellular functions such as microtubule dynamics or DNA replication, leading to apoptosis or programmed cell death” could be more precise at this point. The ADC described carries one topoisomerase I inhibitor whose specific mechanism of action could be described in more detail.
Author Response
This review article is focused in new therapeutic approaches for small cell lung cancer (SCLC). The authors first describe the present therapeutic options and the need for improved treatments given the modest survival benefit of those that are available at the present time. The review is then centred in two new therapeutic approaches, biespecific T-cell engagers (BiTEs) and antibody-drug conjugates (ADCs). The mechanism of action of these new compounds is explained in detail. Later on, the authors described the results obtained for SCLC patients in some clinical trials, one of them for a BiTEs molecule and three for ADC combinations.
The review article is comprehensive and very well presented. It would be of interest for researchers in the field of SCLC and also in that of new cancer therapies involving activation of the immune system.
We appreciate the time and effort that the reviewer dedicated to providing feedback on our manuscript and are grateful for the insightful comments on and valuable improvements to our paper. We have incorporated the suggestions made by the reviewer. Those changes are highlighted within the manuscript.
- Using the same terminology along the manuscript could help to follow the article more easily. For example, biespecific T- cell engagers are named BiTEs in the title and Abstract but BiMAbs in the title of section 3 (line 247). In addition, the group of SCLCs with distant metastasis is named as “extensive disease (ED-SCLC) in the introduction (line 52) and as ES-SCLC at other places along the manuscript, like lines 86 and 427.
- Thank you for pointing this out. We have changed BiMAbs and BiTEs in TCE as suggested by Academic Editor and ED-SCLC in ES-SCLC in introduction.
- The sentence in line 356 should be revised since it states that the ADC generated against B7-H3 binds to DLL3.
- Thank you! We changed DLL3 to B7-H3.
- The general sentence in lines 360-361: “The released drug then disrupts critical cellular functions such as microtubule dynamics or DNA replication, leading to apoptosis or programmed cell death” could be more precise at this point. The ADC described carries one topoisomerase I inhibitor whose specific mechanism of action could be described in more detail.
- We think this is an excellent suggestion. We modified the sentence and focused on topoisomerase I inhibitor mechanism.
Reviewer 2 Report
Comments and Suggestions for Authors
1. Language and Expression
There are instances of unnatural or redundant phrasing throughout the manuscript. For example:
“The structural composition of a conventional BiTE consists of two primary components: scFvs and flexible linker regions.”
This could be more concisely written as:
“Conventional BiTEs comprise two major components: scFvs and flexible linkers.”
I strongly recommend professional language editing prior to submission. In its current form, the manuscript is not suitable for publication.
2. Accuracy of Statements
Some of the conclusive statements in the manuscript require refinement for accuracy. For instance, the opening sentence reads:
“Small cell lung cancer (SCLC) is a malignant epithelial tumour that originates from the lung or, rarely, from extrapulmonary sites (gastrointestinal or genitourinary tract). The main risk factor for its development is tobacco smoking.”
I suggest the following revision for clarity and precision:
“Small cell lung cancer (SCLC) is a high-grade malignant epithelial tumor that arises predominantly in the lung. Rarely, small cell carcinomas with similar histological features may arise from extrapulmonary sites (e.g., gastrointestinal or genitourinary tract), but these are classified as extrapulmonary small cell carcinomas (EPSCC). Tobacco smoking is the primary risk factor for pulmonary SCLC.”
3. Figures and Tables
The number of figures and tables is relatively limited. I recommend including at least two additional visual elements to enhance the manuscript.
While Figures 1 and 2 are helpful for illustrating structural mechanisms, they lack clear source attribution. It is advisable to include annotations such as:
“Figure created with BioRender.com and adapted from [ref].”
Table 1 is informative, but the formatting is quite dense. Improving the layout—such as breaking down each column into more digestible segments—would significantly enhance readability.
4. Insufficient Depth of Analysis
Although multiple clinical trials are cited, the manuscript lacks comparative analysis regarding the differences in mechanisms of action, toxicity profiles, patient selection, and target specificity among the discussed agents.
I recommend adding a dedicated subsection to compare BiTEs and ADCs—e.g., “BiTEs vs ADCs: Therapeutic Scenarios and Challenges”—and to explore the potential correlation between molecular subtypes (such as SCLC-A/N/I) and treatment responses to these novel agents.
5. Conclusions Could Be More Specific
The current conclusion section is rather general and lacks critical insight. For example, phrases like “the results of ongoing studies will further clarify…” are vague.
I suggest expanding this section to provide more concrete perspectives on the clinical prospects of the discussed therapies, as well as possible integration strategies (e.g., combination with ICIs or radiotherapy).
Comments on the Quality of English LanguageThere are instances of unnatural or redundant phrasing throughout the manuscript. I strongly recommend professional language editing prior to submission. In its current form, the manuscript is not suitable for publication.
Author Response
The authors would like to thank the Reviewer for his comments. Care has been taken to improve the work and address their concerns as per the specific comments below.
- Language and Expression
There are instances of unnatural or redundant phrasing throughout the manuscript. For example:
“The structural composition of a conventional BiTE consists of two primary components: scFvs and flexible linker regions.”
This could be more concisely written as:
“Conventional BiTEs comprise two major components: scFvs and flexible linkers.”
I strongly recommend professional language editing prior to submission. In its current form, the manuscript is not suitable for publication.
- As suggested by the reviewer, we have modified some sentences to be more natural and less redundand.
- Accuracy of Statements
Some of the conclusive statements in the manuscript require refinement for accuracy. For instance, the opening sentence reads:
“Small cell lung cancer (SCLC) is a malignant epithelial tumour that originates from the lung or, rarely, from extrapulmonary sites (gastrointestinal or genitourinary tract). The main risk factor for its development is tobacco smoking.”
I suggest the following revision for clarity and precision:
“Small cell lung cancer (SCLC) is a high-grade malignant epithelial tumor that arises predominantly in the lung. Rarely, small cell carcinomas with similar histological features may arise from extrapulmonary sites (e.g., gastrointestinal or genitourinary tract), but these are classified as extrapulmonary small cell carcinomas (EPSCC). Tobacco smoking is the primary risk factor for pulmonary SCLC.”
- Thanks! This has now been corrected as suggested.
- Figures and Tables
The number of figures and tables is relatively limited. I recommend including at least two additional visual elements to enhance the manuscript.
While Figures 1 and 2 are helpful for illustrating structural mechanisms, they lack clear source attribution. It is advisable to include annotations such as:
“Figure created with BioRender.com and adapted from [ref].”
Table 1 is informative, but the formatting is quite dense. Improving the layout—such as breaking down each column into more digestible segments—would significantly enhance readability.
- We added a new table, a new figure and improved the layout of Table 1.
- Insufficient Depth of Analysis
Although multiple clinical trials are cited, the manuscript lacks comparative analysis regarding the differences in mechanisms of action, toxicity profiles, patient selection, and target specificity among the discussed agents.
I recommend adding a dedicated subsection to compare BiTEs and ADCs—e.g., “BiTEs vs ADCs: Therapeutic Scenarios and Challenges”—and to explore the potential correlation between molecular subtypes (such as SCLC-A/N/I) and treatment responses to these novel agents.
- We dedicated a subsection to compare BiTEs and ADCs as raccomended.
- Conclusions Could Be More Specific
The current conclusion section is rather general and lacks critical insight. For example, phrases like “the results of ongoing studies will further clarify…” are vague.
I suggest expanding this section to provide more concrete perspectives on the clinical prospects of the discussed therapies, as well as possible integration strategies (e.g., combination with ICIs or radiotherapy).
- Thank you! We improved the conclusions.
Round 2
Reviewer 2 Report
Comments and Suggestions for Authors
The English writing of the submitted manuscript still requires improvement. I recommend that the authors seek assistance from a professional language editing service for a thorough revision.
Here are some examples of language issues found in the manuscript:
1. Grammatical and Expression Errors
Example:
"chemotherapy representing the solely treatment strategy"
Correction: “solely” should be replaced with “sole.”
Another example:
“The association between cisplatin and etoposide… became a ‘one-size-fits-all’ approach…”
Suggestion: Use more scientific phrasing such as
“…became the standard-of-care regimen…”
2. Redundant or Repetitive Phrasing
Repeated expressions such as “…binds to CD3 on T cells and to DLL3…” appear multiple times and could be streamlined.
Example:
“The scFvs are linked by a short peptide linker…”
The word “link” is repeated unnecessarily and could be simplified.
3. Unnatural or Awkward Expressions
Example:
“on the other [hand], they can deliver new and promising chemotherapeutic agents directly to the tumor…”
More natural phrasing:
“…alternatively, they enable direct tumor delivery of potent chemotherapeutic agents…”
4. Overly Long Sentences That Impede Readability
Example:
“This dual-targeting approach creates a direct link between T cells and tumor cells, thereby facilitating an immune response against cancer.”
Suggested revision:
“By linking T cells and tumor cells, this dual-targeting approach enhances immune-mediated tumor killing.”
5. Typos and Grammar Mistakes
Example:
“transcipion factor” should be “transcription factor”
“treatment e tolerability” should be “treatment and tolerability”
“impact of tarlatamab activity…” should be “impact of tarlatamab on activity…”
Author Response
Dear reviewer, we thank you for your valuable comments. In this revised version, we addressed all the issues raised during the last revision and edited our manuscript in line with your suggestions.

Round 3
Reviewer 2 Report
Comments and Suggestions for Authors
Detailed Suggestions for Minor Revisions (English Version)
1. Language Simplification and Fluency Improvements
Some sentences could be slightly reworded for better conciseness and natural flow:
Original:
"Lung cancer is the second most common cancer in US after prostate cancer and breast cancer in men women, respectively..."
Suggested revision:
"Lung cancer is the second most common cancer in the US, following prostate cancer in men and breast cancer in women."
Original:
"Despite the high overall response rates, this therapeutic strategy failed to provide long-term benefits for most of the patients and many of them still have a dismal prognosis..."
Suggested revision:
"Despite high overall response rates, this strategy failed to deliver long-term benefits for most patients, who continue to face a poor prognosis."
Original:
"DLL3 is a protein that inhibits Notch signaling, is usually localized intracellularly in normal cells, but is abnormally expressed on the surface of SCLC cells..."
Suggested revision:
"DLL3, normally localized intracellularly, inhibits Notch signaling but is aberrantly expressed on the surface of SCLC cells."
2. Typographical Errors or Minor Grammatical Issues
Some minor corrections are recommended:
"men women" âž” Should be "men and women"
"Median overall survival (mOS) was 12.3 months vs 10.3 in standard one" âž” Suggest clarifying as "compared to 10.3 months in the standard arm."
"Median progression-free survival (mPFS) was 5.2 months and 4.3 months" âž” Add "respectively": "5.2 months and 4.3 months, respectively."
Maintain consistency in American English spelling (e.g., "tumor," which is already well used throughout the text).
3. Logical Structure and Transitional Phrases
Adding a few transitional phrases could help improve the logical flow between sections:
Between Section 1.2 (Standard of care) and Section 2 (TCEs and ADCs), a transition sentence could be inserted:
"Given the limited success of traditional chemotherapy and immunotherapy, innovative therapeutic modalities such as TCEs and ADCs have gained increasing attention."
When introducing different therapies (e.g., moving from tarlatamab to sacituzumab govitecan), consider adding bridging sentences like:
"In addition to DLL3-targeted therapies, TROP-2-targeted ADCs have also demonstrated promising efficacy in SCLC."
4. Paragraph Splitting for Better Readability
To improve readability:
Aim to limit each paragraph to around 100–150 words.
Particularly for clinical trial descriptions (such as DeLLphi-301 or TROPiCS-03), the content could be divided into three distinct parts:
Trial design and basic information
Efficacy outcomes (ORR, PFS, OS)
Adverse events (AEs)
For example, the description of the DeLLphi-301 study could be split into three shorter paragraphs.
5. Minor Formatting Adjustments in Figures and Tables
Units such as mPFS and mOS should be consistently expressed throughout tables (e.g., always write "months (mos)" rather than sometimes using "m" or "mos" inconsistently).
Cross-check that all "Table 1," "Table 2," etc., references match between the main text and the figures/tables.
Figure legends could be slightly expanded. Before "Created with BioRender.com," add a brief explanation such as:
"Schematic representation of the TCE structure and mechanism. Created with BioRender.com."
Author Response
- Language Simplification and Fluency Improvements
Some sentences could be slightly reworded for better conciseness and natural flow:
Original:
"Lung cancer is the second most common cancer in US after prostate cancer and breast cancer in men women, respectively..."
Suggested revision:
"Lung cancer is the second most common cancer in the US, following prostate cancer in men and breast cancer in women."
Original:
"Despite the high overall response rates, this therapeutic strategy failed to provide long-term benefits for most of the patients and many of them still have a dismal prognosis..."
Suggested revision:
"Despite high overall response rates, this strategy failed to deliver long-term benefits for most patients, who continue to face a poor prognosis."
Original:
"DLL3 is a protein that inhibits Notch signaling, is usually localized intracellularly in normal cells, but is abnormally expressed on the surface of SCLC cells..."
Suggested revision:
"DLL3, normally localized intracellularly, inhibits Notch signaling but is aberrantly expressed on the surface of SCLC cells."
- As suggested by the reviewer, we have modified some sentences for better conciseness and natural flow.
- Typographical Errors or Minor Grammatical Issues
Some minor corrections are recommended:
"men women" âž” Should be "men and women"
"Median overall survival (mOS) was 12.3 months vs 10.3 in standard one" âž” Suggest clarifying as "compared to 10.3 months in the standard arm."
"Median progression-free survival (mPFS) was 5.2 months and 4.3 months" âž” Add "respectively": "5.2 months and 4.3 months, respectively."
Maintain consistency in American English spelling (e.g., "tumor," which is already well used throughout the text).
- Thanks! We have corrected the sentences as recommended.
- Logical Structure and Transitional Phrases
Adding a few transitional phrases could help improve the logical flow between sections:
Between Section 1.2 (Standard of care) and Section 2 (TCEs and ADCs), a transition sentence could be inserted:
"Given the limited success of traditional chemotherapy and immunotherapy, innovative therapeutic modalities such as TCEs and ADCs have gained increasing attention."
When introducing different therapies (e.g., moving from tarlatamab to sacituzumab govitecan), consider adding bridging sentences like:
"In addition to DLL3-targeted therapies, TROP-2-targeted ADCs have also demonstrated promising efficacy in SCLC."
- We added transitional phrases to improve the logical flow.
- Paragraph Splitting for Better Readability
To improve readability:
Aim to limit each paragraph to around 100–150 words.
Particularly for clinical trial descriptions (such as DeLLphi-301 or TROPiCS-03), the content could be divided into three distinct parts:
Trial design and basic information
Efficacy outcomes (ORR, PFS, OS)
Adverse events (AEs)
For example, the description of the DeLLphi-301 study could be split into three shorter paragraphs.
- We have reduced the words for each paragraph to improve readability.
- Minor Formatting Adjustments in Figures and Tables
Units such as mPFS and mOS should be consistently expressed throughout tables (e.g., always write "months (mos)" rather than sometimes using "m" or "mos" inconsistently).
Cross-check that all "Table 1," "Table 2," etc., references match between the main text and the figures/tables.
Figure legends could be slightly expanded. Before "Created with BioRender.com," add a brief explanation such as:
"Schematic representation of the TCE structure and mechanism. Created with BioRender.com."
- Thanks. We have checked all tables and figures.